# Metabolic Biomarkers in Adults with Type 2 Diabetes: The Role of PPAR-γ2 and PPAR-β/δ Polymorphisms

**DOI:** 10.3390/biom13121791

**Published:** 2023-12-14

**Authors:** Sandra A. Reza-López, Susana González-Gurrola, Oscar O. Morales-Morales, Janette G. Moreno-González, Ana M. Rivas-Gómez, Everardo González-Rodríguez, Verónica Moreno-Brito, Angel Licón-Trillo, Irene Leal-Berumen

**Affiliations:** 1Facultad de Medicina y Ciencias Biomédicas, Universidad Autónoma de Chihuahua, Campus II. Circuito Universitario S/N, Chihuahua 31109, CP, Mexico; sreza@uach.mx (S.A.R.-L.); osmorales@uach.mx (O.O.M.-M.); jgmoreno@uach.mx (J.G.M.-G.); evegonzal@uach.mx (E.G.-R.); vmoreno@uach.mx (V.M.-B.); alicon@uach.mx (A.L.-T.); 2Instituto Mexicano del Seguro Social UMF 33, Avenida Melchor Ocampo y Arroyo de los Perros S/N, Col. El Palomar, Chihuahua 31204, CP, Mexico; susyglzg@hotmail.com (S.G.-G.); or ana.rivas@imss.gob.mx (A.M.R.-G.)

**Keywords:** T2D metabolic biomarkers, PPAR polymorphisms (SNPs), glycemic control, PPARG Pro12Ala, PPARD +294T/C

## Abstract

Glucose and lipid metabolism regulation by the peroxisome proliferator-activated receptors (PPARs) has been extensively reported. However, the role of their polymorphisms remains unclear. Objective: To determine the relation between PPAR-γ2 rs1801282 (Pro12Ala) and PPAR-β/δ rs2016520 (+294T/C) polymorphisms and metabolic biomarkers in adults with type 2 diabetes (T2D). Materials and Methods: We included 314 patients with T2D. Information on anthropometric, fasting plasma glucose (FPG), HbA1c and lipid profile measurements was taken from clinical records. Genomic DNA was obtained from peripheral blood. End-point PCR was used for PPAR-γ2 rs1801282, while for PPAR-β/δ rs2016520 the PCR product was digested with Bsl-I enzyme. Data were compared with parametric or non-parametric tests. Multivariate models were used to adjust for covariates and interaction effects. Results: minor allele frequency was 12.42% for PPAR-γ2 rs1801282-G and 13.85% for PPAR-β/δ rs2016520-C. Both polymorphisms were related to waist circumference; they showed independent effects on HbA1c, while they interacted for FPG; carriers of both PPAR minor alleles had the highest values. Interactions between FPG and polymorphisms were identified in their relation to triglyceride level. Conclusions: PPAR-γ2 rs1801282 and PPAR-β/δ rs2016520 polymorphisms are associated with anthropometric, glucose, and lipid metabolism biomarkers in T2D patients. Further research is required on the molecular mechanisms involved.

## 1. Introduction

Diabetes is a serious public health issue with a 10.5% worldwide prevalence [1]. In Mexico, it reaches about 18.3% in the adult population [2]. Type 2 diabetes (T2D) is the most common form of diabetes, and accounts for over 90% of cases [1]. Numerous and diverse environmental and genetic factors are involved in T2D development. Medical history, serum biomarkers, psychosocial and lifestyle factors, unhealthy diet, adiposity, family history, age, ethnicity, and genetics have all been related to T2D and glycemic dysregulation [1,3,4,5]. 

Diabetes can lead to complications such as cardiovascular disease, nephropathy, or retinopathy [6], which have in turn been related to hyperglycemia, hypertension, and hypercholesterolemia. Decreasing HbA1c may reduce the risk of microvascular complications by 40% [7]. Therefore, glycemic control is a common treatment target for T2D patients. However, several reports indicate that this goal is achieved by less than 50% of patients. Lifestyle variables, medication, and individual characteristics [8,9], as well as genetics have been related to glycemic control. Among the latter, several gene polymorphisms have been associated with the response to medications [10]. However, few studies have reported the relation of polymorphisms and glycemic control by other pathways [11,12]. For instance, poor glycemic control has been observed in carriers of the T allele of rs2241766 SNP in ADIPOQ, a gene that encodes for adiponectin, which is an adipokine related to insulin sensitivity [11]. 

Other genes that play important roles in glucose and lipid metabolic networks are those encoding for the peroxisome-proliferator-activated-receptors (PPAR) [13,14,15]. There are three main PPAR nuclear receptors, each isoform with different tissue distribution and activated by specific ligands with wide physiological functions [16,17]. PPAR-β/δ has been shown to promote glycolysis, glucose absorption, glycogen storage, and gluconeogenesis downregulation in animal models [18,19]; similar to PPAR-α, it is related to fatty acid oxidation, increase serum high-density lipoproteins (HDL) levels with lower plasma triglycerides (TG), and very-low-density lipoproteins (VLDL) [15]. PPAR-γ has three isoforms expressed in different tissues. PPAR-γ contributes to lipid synthesis and storage in white adipose tissue, and plays a role in insulin sensitivity and glucose homeostasis [20,21]. 

Several polymorphisms of PPARG and PPARD genes have been described. PPAR-γ2 rs1801282 is a missense single-nucleotide-polymorphism (SNP) in the B exon on chromosome 3p25.3 (OMIM number 601487), where a cytosine/guanine (C/G) transversion promotes a polypeptide substitution of proline to alanine in the 12th amino acid (Pro12Ala) [22]. This variant has been related to anthropometric measures, insulin resistance and serum lipids levels [23,24]. PPAR-β/δ rs2016520 (+294T/C) is a transition SNP located in the 5’UTR exon 4 region on chromosome 6p21.31 (OMIM number 600409). While it is not associated with T2D risk [25], studies in different populations have reported the association of this PPAR-β/δ SNP with glucose and lipid biomarkers [26,27]; however, studies in subjects with T2D have shown diverse results [25,28]. In this study, we aimed to determine the frequencies of PPAR-γ2 rs1801282 (Pro12Ala) and PPAR-β/δ rs2016520 (+294T/C) polymorphisms and their relation to anthropometric and metabolic biomarkers in adults with T2D from Chihuahua, Northern Mexico.

## 2. Materials and Methods

### 2.1. Study Design and Population

This cross-sectional study included 314 patients diagnosed with T2D by their treating physicians at the Unidad de Medicina Familiar, Clínica 33 of the Instituto Mexicano del Seguro Social (UMF33-IMSS) Chihuahua, Chihuahua, Mexico. Patients were older than 18 years of age, affiliated with UMF33-IMSS Chihuahua, they had a clinical diagnosis for T2D in their file, and were willing to donate a blood sample. Patients who had interest in participating and who signed a written informed consent were included in the study. The study was approved by the institutional ethics research committees with the number R-2020-805-028 and CI-020-19 for the IMSS and the Facultad de Medicina y Ciencias Biomédicas, Universidad Autónoma de Chihuahua, respectively.

### 2.2. Phenotyping and Biochemical Data

Patients’ anthropometric and biochemical data were obtained from their medical records. Anthropometric measures were: height (m), weight (kg), and waist circumference (cm). Body mass index (BMI) was calculated as the weight in kilograms divided by the square of the height in meters (kg/m^2^) and classified as follows: ≤24.9 as normal weight, 25–29.9 overweight, 30–34.9 obesity class I, 35–39.9 obesity class II, and ≥40 obesity class III. Glycemic control was considered with a glycosylated hemoglobin A1c (HbA1c) < 7% according to the Mexican guidelines [29]. The TC/HDL-C index was calculated [30]. 

### 2.3. Genotyping

The genomic DNA was isolated from peripheral blood buffy coats using a Master Pure Epicenter DNA extraction Kit (Madison, Wisconsin, USA) following the manufacturer’s recommendations. Genotyping of PPAR-γ2 rs1801282 (Pro12Ala) gene and the PPAR-β/δ rs2016520 (+294T/C) were performed as previously described [27]. 

The polymerase chain reaction (PCR) conditions used for both SNPs were: an initial 3 min denaturation at 94 °C, followed by 35 cycles of denaturation at 94 °C for 45 s, annealing at 62 °C for 45 s, and extension at 72 °C for 45 s. The final extension step consisted of 5 min at 72 °C (Agilent SureCycler 8800, Santa Clara, CA, USA). 

Electrophoresis in 3.5% agarose gels was used to identify PPAR-γ2 rs1801282 genotypes. The PPAR-β/δ rs2016520 -269-bp-resulting-PCR- product was digested with 5 U fast Bsl-I (Thermo Scientific, Madison, WI, USA) restriction enzyme for 1 h at 37 °C. The resulting fragments and the primers used for genotyping are presented in Table 1.

### 2.4. Statistical Analysis 

The Χ^2^ test verified Hardy–Weinberg equilibrium (H-WE). Quantitative measures were described as means and standard deviation (SD) or median and interquartile range (IQR) for variables with normal or non-normal distribution, respectively. Absolute and relative frequencies are presented for categorical variables. Three inheritance models were analyzed: codominant, dominant, and over-dominant. Anthropometric and biochemical measures were compared by inheritance model with Student’s *t*-test or Wilcoxon’s rank sum test (Mann–Whitney U test); analysis of variance or the Kruskal–Wallis test, followed by Bonferroni’s post hoc test or Dunn’s test, as appropriate. Categorical variables were compared by Pearson’s Χ^2^ test. A *p* < 0.05 value was considered significant.

Multiple linear regression models were used to adjust the relation between biochemical measures as continuous variables (log-transformed when needed to reach normality) and PPAR for variables such as age, sex, age at T2D diagnosis, BMI, and waist circumference as potential confounders and to test interaction effects. Variables with *p* < 0.05 were kept in multivariate analyses. A stratified analysis was conducted when an interaction effect was identified (*p* ≤ 0.10). The normality of residuals, homoscedasticity, and variance inflation factors were tested. Logistic regression was used to analyze the relation between HbA1c as categorical variable (<7% vs. ≥7%) and PPAR SNPs, adjusting for potential confounders as covariates. Stata v. 14.2 (2015, Stata Corp., College Station, TX, USA) was used for statistical analysis.

## 3. Results

### 3.1. Population Characteristics

A total of 314 participants with the following characteristics were included: age of 57.9 ± 10.6 years (mean ± SD); age at the time of T2D diagnosis of 48.9 ± 9.9 years and time from T2D diagnosis of 9 (4–13) years (median and IQR); height of 1.62 ± 0.09 m and weight 81 (72–93). According to BMI, 11.2% (n = 35) showed normal weight, 31.2% (n = 98) were overweight, and 57.6% (n = 181) had obesity (33.1% grade I, 15.9% grade II, and 8.6% grade III). 

Almost 60% (59.9%) of T2D patients had uncontrolled glycaemia, defined as HbA1c ≥ 7%: 70% in those who were diagnosed ≤45 years old, 62% in the age group 45–53, and 47% in patients older than 53 years. Ninety percent of participants’ treatment included biguanides, 33% were treated with insulin, and 32% received dipeptidyl peptidase 4 inhibitors (DPP4); moreover, 19% received sulfonylureas, 7% thiazolidinediones, 6% sodium–glucose cotransporter inhibitors, 3.5% alpha-glucosidase inhibitors, and 1% glucagon-like peptide I (GLP-I) agonists. Thirty-four percent had one glucose-lowering drug prescribed, 44% had two of them, and 22% were using three or more. For lipid control, 73% of participants had one medication; 49% were taking statins, 26% used fibrates, and less than 1% used sterol absorption inhibitors.

### 3.2. PPAR-γ2 rs18 01282 and PPAR-β/δ rs2016520

The genotype frequencies for both PPAR polymorphisms showed H-WE equilibrium (Table 2). 

Anthropometric and biochemical measures are presented in Table 3 by PPAR-γ2 rs1801282 and PPAR-β/δ rs2016520 (dominant model). Carriers of at least one minor allele of PPAR-γ2 rs1801282-G had higher values of waist circumference (*p* = 0.029), FPG (*p* = 0.026), and marginally higher HbA1c (*p* = 0.057), while carriers of at least one minor allele of PPAR-β/δ rs2016520-C had higher TG (*p* = 0.026) and VLDL (*p* = 0.026) and tended to have increased waist circumference (*p* = 0.052), FPG (*p* = 0.080) and HbA1c values (*p* = 0.089). According to genotype (codominant model), FPG was higher in rs1801282-CG compared to the CC homozygous for the major allele (*p* = 0.040); TG was greater in rs2016520-TC compared to the homozygous major allele genotype (*p* = 0.046). Similarly, rs2016520-TC tended to higher HbA1c values (*p* = 0.063), and a trend was observed in waist circumference for the heterozygous of both PPARs (*p* = 0.072 and *p* = 0.103 for the rs1801282-CG and rs2016520-TC, respectively) [see Appendix A]. 

### 3.3. PPAR SNPs and Anthropometric Measures

The relation between waist circumference and the studied SNPs were statistically significant after adjusting for the T2D duration (current age–age at the time of diagnosis). Having at least one minor allele of PPAR-γ2 rs1801282-CG + GG was related to higher value of waist circumference (log cm, β = 0.04, *p* = 0.026), while in those with at least one minor allele of PPAR-β/δ rs2016520-TC + CC the relation with waist circumference was significant (log cm, β = 0.04, *p* = 0.029). BMI (log-transformed) was related to PPAR-β/δ rs2016520-TC + CC (β = 0.07, *p* = 0.015) after adjusting for HbA1c and age. PPAR-γ2 rs1801282-G was not significantly related to BMI (*p* = 0.802).

### 3.4. PPAR SNPs and Glycemic Biomarkers

For FPG as dependent variable, an interaction effect between both polymorphisms was observed (*p* = 0.013). Therefore, the geometric means after adjusting for age at the time of diagnosis are presented for rs2016520 separately (TT vs. TC + CC and TT + CC vs. TC) [Figure 1]. The subgroup carrying at least one minor allele of both PPAR SNPs had the highest FPG values after adjusting for age at the time of T2D diagnosis. Similar results were observed in the over-dominant inheritance model. Including the glucose-lowering drugs in the multivariate model did not modify these results. Among patients with insulin treatment (alone or in combination), those with at least a minor allele of the PPAR-γ2 Pro12Ala had higher FPG than the major allele homozygous (*p* for interaction = 0.063).

Both PPAR SNPs were independently associated with HbA1c values (log transformed) in multivariate analyses after adjusting for BMI category, age at the time of T2D diagnosis, and HDL-C. Carriers of at least one minor allele of PPAR-γ2 rs1801282 had higher values of HbA1c. Similarly, carriers of at least one minor allele of PPAR-β/δ rs2016520 had increased HbA1c (Table 4). Treatment with sulfonylureas, insulin, or IDPP4 was related to HbA1c values (*p* < 0.05); however, including pharmacological treatment did not change the direction of the results. 

HbA1c as categorical variable (<7 vs. ≥7%) was associated with the same variables by logistic regression analysis (PPAR-γ2 rs1801282 CC vs. CG + GG: OR =2.14, _95%_CI 1.01, 4.58, *p* = 0.048); PPAR-β/δ rs2016520 TT vs. TC + CC: OR =2.08, _95%_CI 1.03, 4.21, *p* = 0.041), after adjusting for the same variables as the linear regression model. Among glucose lowering drugs, insulin, IDPP4, and sulfonylureas were associated with uncontrolled glycaemia (OR =5.46, OR =4.40, and OR =2.95, *p* < 0.05, respectively) in the fully adjusted model. No interaction effects between glucose-lowering medications and PPAR SNPs were detected. Similar results were observed in the over-dominant model.

### 3.5. PPAR SNPs and Lipid Biomarkers

For TG, we observed an interaction effect between FPG and PPAR-γ2 and PPAR-β/δ polymorphisms; thus, we present them separately (Table 5). FPG was directly related to TG levels in all groups except those carrying both major alleles of the PPAR-β/δ rs2016520-TT and at least one minor allele of PPAR-γ2 rs1801282-CG + GG. For those carrying at least one minor allele of both SNPs, FPG and BMI explained more than 60% of the variability in TG levels (Adj R^2^ = 0.61).

## 4. Discussion

This study analyzed the frequencies of PPAR-γ2 rs1801282 (Pro12Ala) and PPAR-β/δ rs2016520 (+294T/C) polymorphisms and their association with anthropometric and metabolic biomarkers in adults with T2D from a public health institution in Northern Mexico. An interaction effect between these polymorphisms resulted in increased values of FPG in minor allele carriers. Both SNPs were independently associated with increased HbA1c and modified the relation between FPG and TG values. 

The prevalence of PPAR-γ2 rs1801282 (Pro12Ala) polymorphism varies across populations. According to genotype frequency, several studies have reported 0% homozygous GG in Asian Indians [31], original South African [32], while in Romanian children and Caucasian adults from Poland the frequency has ranged between 2.7 and 3.7% [33,34]. In teenagers from our region’s main ethnic groups, it ranges from 3 to 5%, [27]. In Mexican adults with diabetes, the frequency of the G allele was identified as 13.41% [35], similar to that observed in this study (12.42%). PPAR-β/δ rs2016520 (+294T/C) frequency is variable among populations as well. Reported frequencies range from 28.8 to 44.3% of heterozygous (TC) and 3.09 to 12.9% of CC homozygous [36,37,38] in Russian, Chinese, and Mexican populations. Consistent with previous studies in Mexican teenagers from Northern Mexico [27] reporting 1% of CC and 22% of TC, in this study we identified 1.27% of CC homozygous and 25.16% heterozygous in adults with T2D. 

Gene polymorphisms involved in glucose and lipid regulation influence glycemic control in T2D. A recent systematic review revealed a high prevalence of poor glycemic control, ranging between 45.2% to 93% in different countries [39]. In Mexico, glycemic control prevalence was 39% (i.e., uncontrolled 61%) [40]. Similarly, the prevalence of patients with uncontrolled glycaemia found in the present study was 59.9%. In addition, we observed that carriers of minor alleles of the studied PPAR SNPs were independently associated with higher HbA1c values and increased odds for uncontrolled glycaemia (_adj_OR = 2.14 and _adj_OR = 2.08 for PPAR-γ2 rs1801282 and PPAR-β/δ rs2016520, respectively; *p* < 0.05). In multivariate models, HDL-C levels were inversely associated with glycemic control, consistent with other reports in adults with diabetes from the National Health and Nutrition Examination Survey of the United States [41]. In addition, receiving pharmacological treatments such as insulin, IDPP4, and sulfonylureas was associated with poor control. Similarly, lower HDL-C, longer duration of T2D, and treatment with insulin either alone or in combination with oral hypoglycemic agents have all been associated with poor glycemic control in other studies [8,42]. The unexpected associations may be partially explained by the prescription of insulin and other glucose-lowering treatments being recommended to individuals with uncontrolled glycaemia. According to clinical guidelines, dual therapy would be indicated for those patients with uncontrolled glycaemia receiving monotherapy (usually metformin) and triple or insulin schemes when therapeutic goals are not reached [43]. Because we examined prevalent cases with different disease duration, we speculate that cases with poor control are prescribed with more medication. In this way, patients’ glycemic control status could have influenced the number and type of prescribed medications. 

HbA1c is the main biochemical measure recommended to evaluate glycemic control, reflecting average glucose levels. However, other metabolic biomarkers are assessed in T2D patients as well, such as fasting plasma glucose levels and lipid profile [44]. In our study, PPAR-γ2 rs1801282-CG + GG and PPAR-β/δ rs2016520-TC + CC were associated with FPG, HbA1c, and TG. Similar to our findings, in a Bangladeshi T2D population, carriers of PPAR γ2 rs1801282-CG + GG had higher values of FPG and HbA1c [45], while in a T2D study from a Han Chinese population, heterozygous tended to have higher baseline values of FPG, although the difference did not reach statistical significance. However, the decrease in postprandial glucose and TG in response to pioglitazone was greater than in major allele homozygous [46]. PPAR-γ2 rs1801282-CG + GG minor allele carriers had higher values of TG in subjects with obesity, whereas they were lower in T2D patients from Australian and Russian populations, respectively [47,48]. In contrast, FPG, cholesterol, and TG were lower in patients with the rs1801282-CG genotype compared to the homozygous CC in an Iranian population with coronary artery disease (CAD) and T2D [49]. Of note, the levels of these biomarkers were lower than those found in our study, and given the antecedent of CAD, probably received different pharmacological treatment. 

There are few studies of PPAR-β/δ rs2016520 in T2D. In a study investigating the role of PPARD polymorphism in the response to exenatide treatment in newly diagnosed T2D patients from China, baseline values of BMI were higher in patients with TC and CC than in those with TT genotype, while TG values were higher in minor allele homozygous CC of the PPAR-β/δ rs2016520. In their study, fasting insulin and HOMA-IR in carriers of at least one C allele were lower after 6 months of exenatide treatment. However, in multivariate models the rs2016520 SNP was not significantly associated with HbA1c improvement [50]. In another study, TT major allele homozygous responded to nateglinide with a greater decrease in postprandial glucose and increase in HOMA-B than in subjects with the TC/CC genotype [28]. 

The molecular and physiological impacts of these polymorphisms on glucose metabolism are not fully understood. One of the most studied is the PPAR-γ2 Pro12Ala. Carrying the G allele is associated with higher BMI, waist circumference, and TC than in CC homozygous individuals [51]. The PPAR-γ2 rs1801282-G variant has shown decreased transactivation because of its lower affinity to the responsive element and improved insulin sensitivity [52]. By forming a heterodimer with the Retinoid X receptor, PPARγ may induce the adiponectin gene, binding to PPAR-responsive element in human adiponectin promoter (the PPAR-γ isoform was not specified) [53]. Thus, it is possible that PPAR-γ2 polymorphism could modulate adiponectin expression. In an animal study, Heikkinen et al. (2009) analyzed the phenotype and gene expression of homozygous Ala/Ala compared to Pro/Pro under a normal and high fat diet. Ala/Ala mice were leaner, had better insulin sensitivity, and lived longer than Pro/Pro mice. Gene expression analysis of white adipose tissue, muscle, and liver revealed an upregulation of adiponectin receptor 2 expression in adipose tissue and muscle in Ala/Ala mice and in adiponectin expression in Ala/Ala muscle of mice fed a high fat diet. According to the authors, these findings suggest that adiponectin signaling could be involved in the observed Ala/Ala mice characteristics [54]. Studies in Finnish servicemen showed that the Ala allele was associated with increased adiponectin levels in those with more than 10% weight loss [55]. These studies show the complexity of glucose metabolism regulation and the participation of environmental factors.

Less is known about the effect of the PPARD +294T/C polymorphism. PPAR-β/δ plays a role in fatty acid catabolism and energy homeostasis through cellular-level metabolic pathways [56]. Because PPAR-β/δ is an important regulator of lipid metabolism, most studies have focused on investigating lipid profile according to this SNP. The minor allele C had been shown to increase its transcriptional activity compared to the major allele T; in healthy men, carriers of the CC allele had higher LDL-C [57]. In our study in T2D patients, carriers of the C allele had higher TG values (*p* < 0.05) in the dominant inheritance model. In the same line, recent studies in a Chinese population with diabetes found higher TG levels in CC homozygous at the baseline of a study investigating the effects of exenatide. Regarding glycemic control, the same authors reported lower expression of PPAR-β/δ in liver tissue of *db/db* mice and in an insulin resistance model in HepG2 cells. Their results suggest a role of PPAR-β/δ activation in GLP-1R expression regulation, explaining the response to exenatide [50]. Thus, while the C allele might increase transcriptional activity, this could be modified in part by an insulin resistance or dysregulation in T2D patients. 

Outcomes related to the studied PPAR SNPs vary across populations, by ethnic background, individual characteristics, and by conditions such as obesity or T2D. This can be illustrated by the results of a recent meta-analysis by Li et al. (2022) on PPARG rs1801282. Carriers of the G allele had a higher BMI, waist circumference, or differences in lipid profile in several Asian or African populations, while no association with obesity indexes or lipid profile was found in Caucasians, either European or American. The same authors reported interactions of this polymorphism with gender; whereas male G allele carriers had higher BMI, female carriers had higher values of waist circumference [51]. In the same line, some target genes do not show differences between heterozygous and major allele carriers in subjects with morbid obesity, suggesting an interaction with other characteristics [58]. Lifestyle factors contribute to the high variation observed in glycemic responses among individuals as well as interactions between SNPs. Factors such as diet [54], weight loss [55], exercise [59], and fatty acid levels [60] have been documented to influence or interact with SNPs. Furthermore, gene–gene interactions should be taken into account when assessing the influence of SNPs in metabolic biomarkers. Examining polymorphisms, particularly in patients with poor glycemic control, may be important in clinical practice, as genetic variants may influence the response to pharmacological treatment. Furthermore, the effects of genetic variants on metabolic pathways, independently of those for pharmacological treatment actions, remain to be investigated.

Taken together, our results suggest that PPAR polymorphisms are related to metabolic measures in T2D patients. We have to consider several limitations of this study: first, data on lifestyle variables such as nutrient intake and physical activity were not collected; thus, we were unable to assess their role in conjunction with PPAR polymorphisms. Having information on treatment adherence would also have been desirable. Adherence in T2D patients has been reported in about 47% of T2D patients from a similar population in North Mexico [61]. Although treatment adherence could have influenced metabolic biomarkers, we are not aware of differences by the studied SNP genotypes. Second, we included subjects with variable T2D duration, which may influence biomarker values; however, we adjusted models for either age at the time of diagnosis or duration of the disease. As this was a cross-sectional study, only one-time point was selected. Third, we had sample size limitations when fully evaluating biomarkers in carriers of both minor alleles or with haplotype combinations. For instance, heterozygous PPAR-β/δ rs2016520-TC had the highest values of glucose, HbA1c, and TG, whereas minor allele homozygous had the lowest. However, in the dominant model (TC + CC), the overall effect indicated an increase in such biomarkers because of the small number of subjects with the CC genotype (n = 4). Thus, the sample size for the minor allele homozygous did not allow us to analyze them separately. in addition, the sample size limited certain medication–gene interaction analyses for less commonly prescribed medications. Our study included patients from a public health institution with which about 50% of the population is affiliated [62]; thus, our results can be generalized to populations with similar characteristics. 

## 5. Conclusions

In conclusion, PPAR-γ2 rs1801282 (Pro12Ala) and PPAR-β/δ rs2016520 (+294T/C) polymorphisms were associated with anthropometric, glucose and lipid metabolism biomarkers. While there was an interaction effect between both studied polymorphisms for FPG, they showed independent effects for HbA1c. Furthermore, these polymorphisms modified the relation between FPG and TG in adults with T2D. For the best of our knowledge, this is the first study to report a gene–gene interaction between PPAR-β/δ rs2016520 and PPAR-γ rs1801282 in T2D metabolic biomarkers. Further research is needed to gain insight about the mechanisms involved in SNP-related differences, their impact on overall health, and the interaction between lifestyle components and genetic background. The high prevalence of overweight and diabetes patients in Mexico highlights the importance of strategies for primary prevention as well as for preventing disease complications. 

## Figures and Tables

**Figure 1 biomolecules-13-01791-f001:**
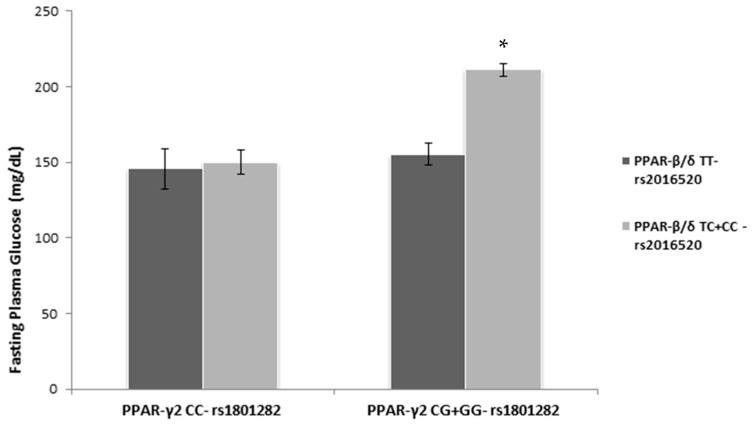
Fasting plasma glucose (geometric means ± SD) by PPAR polymorphisms (inheritance dominant model) adjusted for age at the time of T2D diagnosis. * *p* < 0.05; *p* = 0.013 for the interaction effect. Full data in Appendix A including over-dominant model results.

**Table 1 biomolecules-13-01791-t001:** Genotyping primers for PPAR-γ2 and PPAR-β/δ.

SNP	Sequence	Major Alelle Genotype	Heterozygous Genotype	Minor Alelle Genotype
rs1801282	P1: 5′-GTGTATCAGTGAAGGAATCGCTTTCTTG-3 P2: 5′-TTGTGATATGTTTGCAGACAAGGTATCAG TGAAGGAATCGCTTTGTGC-3′ P3: 5′-TTTCTGTGTTTATTCCCATCTCTCCC-3′	230 bp	230 and 250 bp	250 bp
rs2016520	F: 5′-CATGGTATAGCACTGCAGGAA-3′ R: 5′-CTTCCTCCTGTGGCTGCTC-3′	269 bp	269, 167, and 102 bp	167 and 102 bp

PPAR-γ2 rs1801282: Major homozygous genotype CC (Pro/Pro alleles), heterozygous CG (Pro/Ala alleles), and minor homozygous genotype GG (Ala/Ala alleles). The PPAR-β/δ rs2016520: Major homozygous genotype TT alleles, heterozygous genotype TC alleles, and minor homozygous genotype CC alleles.

**Table 2 biomolecules-13-01791-t002:** Genotype, allele frequencies, and H-WE of PPAR gene polymorphisms.

	PPAR-γ2 rs1801282				PPAR-β/δ rs2016520			
Genotype n (%)	CC 238 (76.77)	CG 67 (21.61)	GG 5 (1.61)	H-WE *p*	TT 231 (73.57)	TC 79 (25.16)	CC 4 (1.27)	H-WE *p*
				0.99				0.63
Allele	C		G		T		C	
n	543		77		541		87	
(%)	(87.58)		(12.42)		(86.15)		(13.85)	

Χ^2^ test.rs2801282-G = minor allele, rs2016520-C = minor allele. Note: four samples did not amplify for PPAR-γ2 rs2801282.

**Table 3 biomolecules-13-01791-t003:** Anthropometric and biochemical measures by PPAR genotypes (dominant model).

Variable	PPAR-γ2 rs1801282			PPAR-β/δ rs2016520		
	CC (n = 238) Md (IQR)	CG + GG (n = 73) Md (IQR)	*p*	TT (n = 231) Md (IQR)	TC + CC (n = 83) Md (IQR)	*p*
BMI (kg/m^2^)	31 (27–35)	31 (28–33)	0.768	31 (27–35)	32 (28–35)	0.238
Waist circumference (cm)	101 (95–110)	106 (99–115)	0.029	100 (95–110)	103 (100–115)	0.052
FPG (mg/dL)	144 (114–179)	157 (127–207)	0.026	145 (117–177)	158 (125–195)	0.089
HbA1c (%)	7 (6–9)	8 (6–10)	0.057	7 (6–9)	8 (6–10)	0.080
TG (mg/dL)	177 (126–235)	181 (138–254)	0.250	174 (124–227)	190 (155–282)	0.026
TC (mg/dL)	194 (164–220)	192 (168–218)	0.838	193 (165–217)	189 (163–226)	0.526
HDL-C (mg/dL)	44 (36–54)	47 (38–56)	0.260	44 (37–54)	46 (37–56)	0.457
LDL-C (mg/d L)	110 (80–134)	99 (73–132)	0.224	109 (79–132)	103 (65–140)	0.671
VLDL (mg/dL)	35 (25–47)	36 (28–51)	0.250	35 (25–45)	38 (31–56)	0.026
TC/HDL-C index	4 (3–5)	4 (3–5)	0.303	4 (3–5)	4 (3–5)	0.964

BMI: body mass index; FPG: fasting plasma glucose; TG: triglycerides; TC: total cholesterol; HDL-C: high-density lipoprotein; LDL-C: low-density lipoprotein; Md: median; IQR: interquartile range.

**Table 4 biomolecules-13-01791-t004:** Multivariate analysis of the relation between PPAR polymorphisms and HbA1c (log-transformed).

Variable	Dominant Model n = 213	Over-Dominant Model n = 213
	β (_95%_CI) *p*	β (_95%_CI) *p*
PPAR-γ2 rs1801282	0.09 (0.02, 0.16) ^a^ 0.011	0.08 (0.01, 0.16) ^c^ 0.020
PPAR-β/δ rs2016520	0.10 (0.03, 0.16) ^b^ 0.005	0.10 (0.04, 0.17) ^d^ 0.003
Body mass index ^e^		
Overweight	−0.11 (−0.21, −0.006) 0.038	−0.11 (−0.21, −0.005) 0.041
Obesity grade I	−0.15 (−0.25, −0.05) 0.004	−0.15 (−0.25, −0.04) 0.005
Obesity grade II	−0.11 (−0.22, 0.01) 0.064	−0.11 (−0.22, 0.01) 0.066
Obesity grade III	−0.18 (−0.31, −0.05) 0.005	−0.18 (−0.31, −0.05) 0.006
Age at the time ofT2D diagnosis (years)	−0.01 (−0.01, −0.004) <0.001	−0.01 (−0.01, −0.004) <0.001
HDL-C (mg/dL)	−0.003 (−0.005, −0.0004) 0.020	−0.003 (−0.005, −0.0003) 0.024

^a^ reference CC, ^b^ reference TT, ^c^ reference CC + GG, ^d^ reference TT + CC, ^e^ reference: normal weight. _95%_CI; 95% confidence interval. Adj R^2^ = 0.16 in both models.

**Table 5 biomolecules-13-01791-t005:** Relation between triglycerides (log mg/dL) and fasting plasma glucose by PPAR polymorphisms (inheritance dominant model).

Variable	PPAR-γ2 rs1801282-CC & PPAR-β/δ rs2016520-TT n = 168	PPAR-γ2 rs1801282-CG + GG & PPAR-β/δ rs2016520-TT n = 61	PPAR-γ2 rs1801282-CC & PPAR-β/δ rs2016520-TC + CC n = 50	PPAR-γ2 rs1801282-CG + GG & PPAR-β/δ rs2016520-TC + CC n = 16
	β (_95%_CI) *p*	β (_95%_CI) *p*	β (_95%_CI) *p*	β (_95%_CI) *p*
Body mass index				
Overweight	0.23 (0.013, 0.454) 0.038	0.31 (−0.457, 1.072) 0.422	0.44 (−0.008, 0.890) 0.054	0.49 (−0.257, 1.242) 0.174
Obesity class I	0.26 (0.038, 0.474) 0.022	0.27 (−0.500, 1.048) 0.479	0.24 (−0.205, 0.690) 0.282	0.93 (0.263, 1.598) 0.011
Obesity class II	0.28 (0.030, 0.521) 0.028	0.17 (−0.658, 1.004) 0.677	0.18 (−0.307, 0.672) 0.458	0.99 (0.247, 1.726) 0.014
Obesity class III	0.19 (−0.093, 0.470) 0.187	−0.03 (−0.970, 0.914) 0.952	0.52 (−0.108, 1.144) 0.103	0.18 (−0.577, 0.946) 0.600
FPG (mg/dL)	0.002 (0.00, 0.003) 0.005	0.00 (−0.002, 0.002) 0.983	0.004 (0.002, 0.005) <0.001	0.002 (0.00, 0.004) 0.033
Adj R^2^	0.06	0.00	0.25	0.61

FPG: fasting plasma glucose; _95%_CI: 95% confidence interval.

## Data Availability

Data are publicly unavailable due to privacy and ethical restrictions. However, they could be made available through a request letter to the corresponding author previous to the ethics committee authorization.

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
