# Peer review of "Metabolic Biomarkers in Adults with Type 2 Diabetes: The Role of PPAR-γ2 and PPAR-β/δ Polymorphisms"

_biomolecules, 2023, doi:10.3390/biom13121791_

Round 1

Reviewer 1 Report

Comments and Suggestions for Authors

Reviewer comments and suggestions

The objective of this study was to determine the relation between PPAR-γ2 rs1801282 (Pro12Ala) and PPAR-β/δ rs2016520 (+294T/C) polymorphisms and metabolic biomarkers in adults with type 2 diabetes (T2D). The study recruited 314 patients with T2D. The study result noted PPAR-γ2 rs1801282-G minor allele frequency was 12.42% and 13.85% for the PPAR-β/δ rs2016520-C. Both polymorphisms were related to waist circumference; they showed independent effects on HbA1c, while for FPG they interacted; carriers of both PPAR minor alleles had the highest values. Hence the study revealed PPAR-γ2 rs1801282 and PPAR-β/δ rs2016520 polymorphisms were associated with anthropometric, glucose and lipid metabolism biomarkers in patients with T2D. 

Overall, the manuscript was well written. However, a few major concerns/comments needed to be explained or modified.

  1. Lines 42-43 The line needs a specific references
  2. Line 48 The authors could start with a better introduction to polymorphisms.
  3. Lines 50 directly comes to PPAR is also not good way. they can discuss the therapeutics as a basic point and then start with assocations of other or links
  4. Line 57 which three isoform the authors was talking about need to explain otherwise delete it
  5. Comments for Genotyping section The author can make a table for this for better llok and understanding, just my suggestion
  6. Line 149-151 Did the author's categories based on the different therapies or its not required
  7. Line 262-263 Please add more in this paragraph about the novelty they obtained
  8. Line 282-283 Please add up the table or figure information in the text
  9. Line 285 What would be the possible reason for this?
  10. Line 316-317 How did the authors discuss this finding with there and also what was the reason the authors mentioned in the study?
  11. Line 326-327 The point needs to be explained well (adiponectin signaling)
  12. Line 335 ( 26 and 46) The study needs to be discussed
  13. Line 338-339 Adding references does not clarify anything

Author Response

Reviewer 1

Comments

Response

Thank you for your comments. They have been very useful to improve this manuscript.

1. Lines 42-43 The line needs a specific references

We have added a specific reference regarding diabetes complications.

Line 43 reference is the same as that for following sentence.

2. Line 48 The authors could start with a better introduction to polymorphisms.

We have improved this section as follows:

Lifestyle variables, medication, and individual characteristics, such as genetics, have been related to glycemic control [8,9]. Among the latter, several gene polymorphisms have been associated with the response to medications [10]. However, few studies have reported the relation of polymorphisms and glycemic control by other pathways [11,12]. For instance, it has been observed a poor glycemic control in carriers of the T allele of rs2241766 SNP in ADIPOQ, a gene that encodes for adiponectin, an adipokine related to insulin sensitivity [11].

Other genes that play important roles in glucose and lipid metabolic networks are the peroxisome-proliferator-activated-receptors (PPAR) [13–15].

3. Lines 50 directly comes to PPAR is also not good way. they can discuss the therapeutics as a basic point and then start with associations of other or links

Changes in the previous paragraph and in the beginning of the next one may offer a better flow.

4. Line 57 which three isoform the authors was talking about need to explain otherwise delete it

We removed it, as our work relates to the PPARG isoform 2 only

5. Comments for Genotyping section The author can make a table for this for better look and understanding, just my suggestion

Thank you for the suggestion. We added a table with information about the primer sequence for genotyping (Table 1) and made the corresponding adjustments in the text.

6. Line 149-151 Did the author's categories based on the different therapies or its not required

These lines describe the patients' pharmacological treatment at the study time. We have described them in the same sentence.

Since patients had different treatment schemes and combinations, we grouped them by the number of medications that they were taking at the study time.

7. Line 262-263 Please add more in this paragraph about the novelty they obtained

We have added the main findings to the discussion opening as follows:

This study analyzed the frequencies of PPAR-γ2 rs1801282 (Pro12Ala) and PPAR-β/δ rs2016520 (+294T/C) polymorphisms and their association with anthropometric and metabolic biomarkers in adults with T2D from a public health institution of the North of Mexico. An interaction effect between these polymorphisms resulted in increased values of FPG in minor allele carriers. Both SNPs were independently associated with increased HbA1c and modified the relation between FPG and TG values.

8. Line 282-283 Please add up the table or figure information in the text

The information on glycemic control prevalence in our study (59.9%) was added in the discussion.  We also included the adjusted odds ratio for the association between the studied polymorphisms and uncontrolled glycaemia.

9. Line 285 What would be the possible reason for this?

We provide some explanation at the end of the paragraph, as follows:

These unexpected associations may be partially explained because the prescription of insulin and other glucose-lowering treatments is recommended to individuals with uncontrolled glycaemia. According to clinical guidelines, dual therapy would be indicated for those patients with uncontrolled glycaemia receiving monotherapy (usually metformin) and triple or insulin schemes when therapeutic goals are not reached [43]. Since we examined prevalent cases, with different disease duration, we speculate that cases with poor control are prescribed with more medication. Then, patients' glycemic control status could have influenced the number and type of prescribed medications.

10. Line 316-317 How did the authors discuss this finding with there and also what was the reason the authors mentioned in the study?

We mentioned the study as an example of how response among genotypes also varies by disease and population. We have removed it, as it may be confusing.

11. Line 326-327 The point needs to be explained well (adiponectin signaling)

We have modified these lines as follows:

PPARγ, forming a heterodimer with Retinoid X receptor, is able to induce adiponectin transcription, binding a PPAR-responsive element in human adiponectin promoter (the PPAR-γ isoform was not specified) [53]. Then, it is possible that PPAR-γ2 polymorphism could modulate adiponectin expression. Heikkinen et al., (2009) in an animal study, ana-lyzed the phenotype and gene expression of the homozygous Ala/Ala, compared to the Pro/Pro, under a normal and high fat diet. Ala/Ala mice were leaner, with better insulin sensitivity, and live longer than Pro/Pro mice. Gene expression analysis of white adipose tissue, muscle, and liver revealed an upregulation of adiponectin receptor 2 expression in adipose tissue and muscle in Ala/Ala mice and in adiponectin expression in Ala/Ala muscle of mice fed a high fat diet. According to the authors, these findings suggest that adiponectin signaling could be involved in the observed Ala/Ala mice characteristics [54]. Studies in Finnish servicemen have shown that Ala allele associates with increased adiponectin levels in those with more than 10% weight loss [55]. The studies show the complexity of glucose metabolism regulation and the participation of environmental factors.

12. Line 335 ( 26 and 46) The study needs to be discussed

We included further information as follows:

In our study in T2D patients, carriers of the C allele had higher TG values (p<0.05) in the dominant inheritance model. In the same line, recent studies in Chinese population with diabetes found higher TG levels in CC homozygous at the baseline of a study investigating the effects of exenatide. Regarding glycemic control, the same authors reported lower expression of PPAR-β/δ liver tissue of db/db mice and in an insulin resistance model in HepG2 cells. Their results suggest a role of PPAR-β/δ activation on increasing GLP-1R expression, thus explaining the response to exenatide [50]. Then, while the C allele might increase transcriptional activity, this could be modified, in part by an insulin resistance or dysregulation in T2D patients.

The study of Wang et. al was discussed in previous paragraphs

13. Line 338-339 Adding references does not clarify anything

We wanted to make the point that we should expect high variation among glycemic responses, due to several factors that should be taken into account. We have clarified this, as follows:

Outcomes related with the studied PPAR SNPs vary across populations, by ethnic background, individual characteristics, and by conditions such as obesity or T2D. This can be illustrated by the results of a recent metaanalysis by Li et al (2022) on PPPARG rs1801282. Carriers of the G allele had higher BMI, waist circumference or differences in lipid profile in several Asian or African populations; while no association with obesity indexes or lipid profile were found in Caucasian, either European or American. The authors also reported interactions of this polymorphism with gender. Whereas male G allele carriers had higher BMI, female carriers had higher values of waist circumference [51]. In the same line, some of the studied SNPs target genes are not different between heterozygous and major allele carriers in subjects with morbid obesity, suggesting an interaction with other characteristics [58]. Lifestyle factors such as diet and physical activity, contribute to the high variation observed in glycemic responses among individuals and interactions between SNPs and factors such as diet [54], weight loss [55], exercise [59], or fatty acid levels [60] have been documented. Furthermore, gene-gene interactions should be taken into account in assessing the influence of SNPs in metabolic biomarkers.

Reviewer 2 Report

Comments and Suggestions for Authors

The authors analyzed the frequencies of specific polymorphisms, PPAR-γ2 rs1801282 (Pro12Ala), and PPAR-β/δ rs2016520 (+294T/C), and their association with anthropometric and metabolic biomarkers in adults with type 2 diabetes (T2D). While the manuscript is well written, there are few major issues that require the authors’ attention:

The study's limitation of a limited sample size is a significant concern. I would recommend the authors clarify the implications of the limited sample on the generalizability of findings or increase the sample size. I would also recommend authors provide information on the duration of medication use for glucose and lipid control. This could be particularly important when interpreting the associations between genetic polymorphisms and treatment outcomes. Authors should also include participants’ adherence to prescribed medications.

For lines 165-174: I would like the authors to discuss the clinical relevance of the observed associations. For instance, if the differences in anthropometric or biochemical measures are statistically significant, explain the potential clinical implications for individuals carrying specific genetic polymorphisms.

While section 3.2 mentions higher values for various measures in carriers of minor alleles, it would be helpful to explicitly state whether these differences are statistically significant. Adding p-values for each association would strengthen the interpretation of the results.

I would recommend authors to mention that no interaction effects between glucose-lowering medications and PPAR SNPs were detected. This adds important information about the potential moderating effects of medications on the genetic associations.

I think the authors should clarify whether the reported associations are adjusted for potential confounding variables. Mentioning the adjustment for factors such as age, BMI category, and HDL-C is important for the interpretation of the results.

Since the study mentions the inclusion of glucose-lowering drugs in the multivariate model, authors should consider providing insights into how these drugs may influence the observed associations. Discuss whether the impact of the drugs is consistent across different genotypes.

Author Response

Reviewer 2 comments

Response

Thank you for your comments. They have been very useful to improve this manuscript.

The study's limitation of a limited sample size is a significant concern. I would recommend the authors clarify the implications of the limited sample on the generalizability of findings or increase the sample size

We agree with the reviewer. Unfortunately, at this time we are unable to increase the sample size. We address these limitations and generalizability of our findings in the discussion:

---we have sample size limitations to fully evaluate biomarkers in carriers of both minor alleles or with haplotype combinations. For instance, heterozygous PPAR-β/δ rs2016520-TC had the highest values of glucose, HbA1c and TG, whereas the minor allele homozygous had the lowest. However, in the dominant model (TC+CC), the overall effect indicated an increase in such biomarkers, because of the small number of subjects with the CC genotype (n=4). Then, the sample size for the minor allele homozygous did not allow us to analyze them separately. Sample size also limited some medication x gene interaction analyses, for less commonly prescribed medications. Our study included patients from a public health institution, which affiliates about 50% of the population [62], thus our results can be generalized to populations with similar characteristics.

I would also recommend authors provide information on the duration of medication use for glucose and lipid control. This could be particularly important when interpreting the associations between genetic polymorphisms and treatment outcomes. Authors should also include participants’ adherence to prescribed medications

We agree with the reviewer. It would be desirable to have data on the duration of medication use and treatment adherence. Since this is a cross-sectional study, only a time point was selected.

We have to consider several limitations of the study: first, data on lifestyle variables, such as nutrient intake and physical activity was not collected, thus we were unable to assess their role in conjunction with PPAR polymorphisms; having information on treatment adherence would have been also desirable. Adherence in T2D patients, has been reported in about 47% of T2D patients from a similar population in North Mexico [61]. Although treatment adherence could have influenced metabolic biomarkers, we are not aware of differences by the studied SNP genotypes. Second, we included subjects with variable T2D duration, which may influence biomarker values; however, we adjusted models for either age at the time of diagnosis or duration of the disease. As a cross-sectional study, only a time point was selected.

For lines 165-174: I would like the authors to discuss the clinical relevance of the observed associations. For instance, if the differences in anthropometric or biochemical measures are statistically significant, explain the potential clinical implications for individuals carrying specific genetic polymorphisms.

Examining polymorphisms, particularly in patients with poor glycemic control, may be important in clinical practice, as genetic variants may influence the response to pharmacological treatment. Furthermore, the effects of genetic variants on metabolic pathways, independently of those for pharmacological treatment actions, are still to be investigated.

While section 3.2 mentions higher values for various measures in carriers of minor alleles, it would be helpful to explicitly state whether these differences are statistically significant. Adding p-values for each association would strengthen the interpretation of the results.

We have added p-values as recommended

I would recommend authors to mention that no interaction effects between glucose-lowering medications and PPAR SNPs were detected. This adds important information

We agree with the reviewer.

Interaction effects were analyzed. In gene x medication interaction, the analysis was limited to the most commonly prescribed medications, because of limited sample size.

No interaction effects were detected between the studied SNPs and medications for HbA1c (continuous or categorical).

Among patients with insulin treatment, those with at least a minor allele of the PPAR-γ2 Pro12Ala, had higher FPG than the major allele homozygous (p for interaction =0.063).

This information is included in Section 3.4

I think the authors should clarify whether the reported associations are adjusted for potential confounding variables. Mentioning the adjustment for factors such as age, BMI category, and HDL-C is important for the interpretation of the results.

We agree with the reviewer. Indeed, we conducted multiple linear or logistic regression adjusting for potential confounders. Only those variables with p<0.05 were kept in multivariate models.

This information is included in the statistical analysis section and in the Figure 1 legend.

Results of the multivariate relation between PPAR SNPs and HbA1c and TG and the potential confounders included as covariates in the multiple linear regression models are presented in tables 4 & 5.

For uncontrolled glycaemia, we used a multiple logistic regression model, then we present the adjusted odds ratios in the text (section 3.4., third paragraph)

Round 2

Reviewer 2 Report

Comments and Suggestions for Authors

The manuscript reads fine now.